# "So sometimes, it looks like it's a neglected ward": Health worker perspectives on implementing kangaroo mother care in southern Malawi

**Mai-Lei Woo Kinshella**[1]*, **Sangwani Salimu**[2], **Brandina Chiwaya**[2], **Felix Chikoti**[2], **Lusungu Chirambo**[2], **Ephrida Mwaungulu**[2], **Mwai Banda**[2], **Laura Newberry**[2], **Jenala Njirammadzi**[2], **Tamanda Hiwa**[2], **Marianne Vidler**[1], **Elizabeth M. Molyneux**[2], **Queen Dube**[3], **Joseph Mfutso-Bengo**[4,5], **David M. Goldfarb**[6], **Kondwani Kawaza**[2,3], **Alinane Linda Nyondo-Mipando**[4]

**1** Department of Obstetrics and Gynaecology, BC Children's and Women's Hospital and University of British Columbia, Vancouver, Canada, **2** Department of Pediatrics and Child Health, College of Medicine, University of Malawi, Blantyre, Malawi, **3** Queen Elizabeth Central Hospital, Pediatrics, Blantyre, Malawi, **4** Department of Health Systems and Policy, School of Public Health and Family Medicine, College of Medicine, University of Malawi, Blantyre, Malawi, **5** Center of Bioethics for Eastern & Southern Africa (CEBESA), Blantyre, Malawi, **6** Department of Pathology and Laboratory Medicine, BC Children's and Women's Hospital and University of British Columbia, Vancouver, Canada

* maggie.kinshella@cw.bc.ca

**Data Availability Statement:** Due to the nature of qualitative transcripts containing potentially directly and indirectly identifying data, the datasets

## Abstract

### Introduction

Kangaroo mother care (KMC) involves continuous skin-to-skin contact of baby on mother's chest to provide warmth, frequent breastfeeding, recognizing danger signs of illness, and early discharge. Though KMC is safe, effective and recommended by the World Health Organization, implementation remains limited in practice. The objective of this study is to understand barriers and facilitators to KMC practice at tertiary and secondary health facilities in southern Malawi from the perspective of health workers.

### Methods

This study is part of the "Integrating a neonatal healthcare package for Malawi" project in the Innovating for Maternal and Child Health in Africa initiative. In-depth interviews were conducted between May-Aug 2019 with a purposively drawn sample of service providers and supervisors working in newborn health at a large tertiary hospital and three district-level hospitals in southern Malawi. Data were analyzed using a thematic approach using NVivo 12 software (QSR International, Melbourne, Australia).

### Findings

A total of 27 nurses, clinical officers, paediatricians and district health management officials were interviewed. Staff attitudes, inadequate resources and reliance on families emerged as key themes. Health workers from Malawi described KMC practice positively as a low-

analyzed during the current study are not publically available to protect participant confidentiality (ethic committees: University of Malawi College of Medicine (P.08/15/1783) and the University of British Columbia (H15-01463-A003). For data inquiries, please contact COMREC Chairperson (comrec@medcol.mw).

**Funding:** MWK, ALNM, KK, QD and DG were funded by the Canadian International Development Research Centre (IDRC), Global Affairs Canada (GAC) and the Canadian Institutes for Health Research (CIHR). Project ID is 108030. The funders had no role in study design, data collection and analysis, decision to publish, or preparation of the manuscript.

**Competing interests:** The authors have declared that no competing interests exist.

cost, low-technology solution appropriate for resource-constrained health settings. However, staff perceptions that KMC babies were clinically stable was associated with lower prioritization in care and poor monitoring practices. Neglect of the KMC ward by medical staff, inadequate staffing and reliance on caregivers for supplies were associated with women self-discharging early.

## Conclusion

Though routine uptake of KMC was policy for stable low birthweight and preterm infants in the four hospitals, there were gaps in monitoring and maintenance of practice. While conceptualized as a low-cost intervention, sustainable implementation requires investments in technologies, staffing and hospital provisioning of basic supplies such as food, bedding, and KMC wraps. Strengthening hospital capacities to support KMC is needed as part of a continuum of care for premature infants.

## Introduction

Kangaroo mother care (KMC) is vital to the survival, well-being and development of small and premature infants; as such, it is an important intervention towards the realization of Sustainable Development Goal 3.2 to end preventable neonatal deaths [1]. Preterm birth is the leading cause of neonatal death around the world, accounting for 36% of neonatal deaths [2] and hospital-based KMC has the potential to avert 50% of preterm deaths [3]. Focusing on small or sick newborns with interventions such as KMC to reduce infection risk, thermal stress and poor nutrition are estimated to prevent nearly 600,000 newborn deaths per year [3].

KMC involves continuous skin-to-skin contact on the mother's chest to keep the baby warm, support for frequent feeding/exclusive breastfeeding, recognizing danger signs of illness, and early discharge. KMC is an effective and safe alternative to incubator care for low birthweight infants in resource-limited health settings with clear benefits in the reduction of neonatal mortality, neonatal sepsis, hypothermia, hypoglycemia and risk of hospital readmission in comparison with conventional care [4, 5]. KMC is also associated with increased exclusive breastfeeding [5]. Developed initially in 1978 by Colombian pediatrician Edgar Rey as a technologically simple solution to a shortage of incubators, KMC is now considered to be a universal best-care tool in both high- and low-resource settings [1, 6]. Kangaroo mother care is recommended by the World Health Organization (WHO) for the routine care of low birthweight babies weighing 2000g or less at birth, initiating as soon as newborns are clinically stable [7]. Yet in spite of over four decades of evidence and WHO recommendations, there are gaps in its routine uptake at health facilities [1, 8, 9].

Malawi has is a neonatal mortality rate of 27/1000 live births and an estimated one in ten babies are born premature (10.5%) [10, 11]. Though Malawi was one of the early adopters of KMC, piloting the intervention in 1999 and integrated into national policy for routine care in 2005, KMC has not been adequately scaled up across the country [12]. While promoted as a low-cost, low-technology solution appropriate for low-resource health settings, there is a need to understand health system contexts and health worker engagement. The objective of this study is to understand barriers and facilitators to KMC practice at tertiary and secondary health facilities from southern Malawi from the perspective of service providers and supervisors.

## Methodology

### Design

This study is part of the larger project, "Integrating a neonatal healthcare package for Malawi" which seeks to inform the scale-up of low-cost and locally appropriate innovations to improve newborn care at low-resource health facilities. The project is a part of the Innovating for Maternal and Child Health in Africa (IMCHA) initiative by the Canadian International Development Research Centre, Global Affairs Canada and the Canadian Institutes for Health Research. We conducted an exploratory qualitative study following a phenomenological approach to explore the perspectives and experiences of health care workers that work with KMC in their facilities. The study is reported based on the "Consolidated criteria for reporting qualitative research" (COREQ) [13] (S1 File). The study received ethics approval from the University of Malawi College of Medicine (P.08/15/1783) and the University of British Columbia (H15-01463-A003).

### Research setting

The study was conducted at a tertiary-level central hospital and three secondary-level district hospitals in southern Malawi. The two government district hospitals and one private not-for-profit mission hospital included in this study serve as regional referral centres and represent the highest level of care available in their districts. Essential services, including maternal and child healthcare, are provided free of charge to patients in all four study sites.

### Recruitment and selection

The sample was purposively drawn to include health workers and supervisors working in newborn health. At the tertiary facility, we recruited nurses, clinicians and pediatricians. At the secondary facilities, we recruited nurses, clinicians and district health management (district health officers, district medical officers, district nursing officers) who oversaw the delivery of health services. Health workers were approached in person or by phone and asked for an interview after being briefed on the study. Based on the limited number of health workers in neonatal units, a sample size of 5–10 per site was estimated to reach data saturation with a variety of perspectives.

### Data collection and analysis

Interviews took place between May and August 2019 with a semi-structured topic guide (S2 and S3 Files). Four research nurses (BC, FC, LC, EM, Diploma in Nursing & Midwifery, 3 females and 1 male) and a public health specialist (SS, Masters in Public Health, female) were hired as part of the IMCHA study and underwent a three-day intensive training in qualitative research methods. Participants did not know the interviewers prior to the study. Face-to-face interviews were conducted in a secluded place within the facilities after participants provided written informed consent. The 30–60 minute interviews were conducted in English or Chichewa, the major local language in Malawi. No health workers refused to participate and no repeat interviews were conducted.

We employed a thematic approach in analyzing the data. The interview guide was piloted with nurses at the central hospital to refine questions and preliminary analysis to develop the coding framework for analysis. The coding framework included training, initiation, monitoring, perceptions of KMC among health workers, health worker perceptions of caregiver attitudes and engagement, overall challenges and supports of KMC practice. Codes emerged inductively from the pilot data as well as deductively from our research questions. Due to the

exploratory approach of the qualitative research, data obtained from the pilot interview was included in full analyses. Field notes and interviews were recorded with permission and translated verbatim. Participants were given opportunity to review transcripts, though no participants opted to view. Data was analyzed on NVivo 12 software (QSR International, Melbourne, Australia) (MWK, MA in Medical Anthropology, and ALMN, PhD in Health Systems and Policy). Confidentiality of the data collected was ensured through separation of the interviews, demographics and signed consent forms, storing data in a secure, locked filing cabinet and de-identifying participants using codes and aggregating demographic features.

## Results

A total of 27 health workers and supervisors were interviewed. Nine were from the central hospital, including five nurses, two clinical officers and two paediatricians. Seventeen were interviewed from district hospitals, including nine nurses, two clinical officers and seven from the district health management team.

### Staff attitudes

Health workers in general had positive perceptions of KMC, which was identified as a facilitator of practice. They felt KMC is a good practice because it helps to save lives. Health workers described that KMC promotes mother and infant bonding and frequent feeding. Health workers shared that KMC was valuable as a low-cost, low-tech solution in their resource-constrained health settings. Some also highlighted that KMC reduced workload and increased protection from complications and infections as mothers closely monitored their infants and there were fewer people handling the vulnerable neonate.

> "It is something very cheap. It's cheap and. . .it's the most effective way of helping our premature babies to gain weight so fast!" District hospital nurse

> "If the mother put the baby on kangaroo, it's easy for the mother to know that the baby is sick or not because the mother sees the baby. . .All eyes of the mother is on her baby. . .. The mother [can quickly see] "my baby is not breathing". . .so the baby is protected from any danger" Tertiary hospital nurse

> "KMC has helped us in various ways. It has reduced workload because this low birth weight baby is at risk of developing different problems. This means that if there was no KMC, midwives would have been busy attending to such problems. KMC has simplified our life because such things that were supposed to be done by us, mothers are doing them." Tertiary hospital nurse

Though KMC babies were considered vulnerable as preterm and low birthweight infants, health workers also perceived them as in stable condition. Health workers highlighted that only infants that were clinically stable were eligible for KMC according to guidelines at their hospitals. Birth asphyxia and respiratory distress were described as complications that preterm and low birthweight newborns frequently experienced requiring critical care before starting KMC. Additionally, delays in initiation may also occur while waiting for the mother's condition to stabilize, such as recovery after a caesarean delivery. With an emphasis on stable health of mother and infant prior to initiating KMC, health workers shared that it was a common attitude that KMC babies were a low priority in comparison to those in critical care. This was associated with irregular monitoring of the KMC ward and staff expected the mother to call for help if complications arose.

"Attitude. . . yah sometimes, you know, committing a crime by omission. . . some people will just say, "aahh these KMC babies are stable, [so] let me concentrate on this one" and yet, you forget that something can happen and the mom might not notice" District medical officer

"Aah. . .there is no constant monitoring of the babies in the ward. It might happen that only the babies are being monitored for two hours in a day or once a day. . . then clinicians don't do the ward round for two to three days in KMC. You just need to pull them to go there. . . so sometimes, it looks like it's a neglected ward" District hospital nurse

## Inadequate resources

Inadequate staffing was reported to challenge both KMC initiation and monitoring. Initiation required time to counsel mothers on appropriate practice and introduce her to the KMC ward. At the district hospitals, there were fewer staff on duty at night in the nursery. At the tertiary hospital where there are specific staff for the KMC ward, the KMC ward is unstaffed at night and there are fewer nurses in the critical care nursery. The need to care for sick babies first and nurses pulled into labour and delivery wards, especially at district hospitals, contributed to delays in initiation and monitoring. Health workers described an imbalance between workload and a number of staff that challenged their ability for regular monitoring.

"I had this situation where there was neonate born premature and the neonate was initiated on KMC and the baby was transferred to kangaroo ward. During the night, I don't know what happened, the mother came from kangaroo ward and presented her baby to me to assess the baby. When I assessed the baby, the baby was already dead. I can say that our KMC ward has no nurses on duty during the night, this baby was not monitored properly and the mother seem to be lacking knowledge on how to take care of the baby. I believe if there was proper monitor the baby would have been attended to before she had died." Tertiary hospital nurse

"During the night, we work [as] three people: one postnatal staff, one in the labour ward and one in the nursery. . .[but] we can see that one staff in the labour ward cannot work alone, so they rely on postnatal and nursery and they mix. Let's say, labour ward, there are two or three staff. . .that means the nursery is on their own, postnatal alone. . . these things make our monitoring poor" District hospital nurse

Additionally, effective monitoring of vital signs and clinical indicators while practicing KMC was challenged by lack of equipment, such as a weighing scale, thermometer and glucometer.

"Lack of equipment make monitoring difficult, for example if there are no thermometers for measuring temperature it becomes difficult, and also glucometer and other equipment if they are not available it makes monitoring a challenge" District hospital nurse

Health workers reported that sharing roles and responsibilities as a team between nurses and clinicians supported effective KMC practice as well as appropriate infrastructure and supplies. Health workers noted that having a dedicated KMC space with proper beds was a facilitator to KMC practice at their health facilities.

"I think we, since we have got better space now, better place the mum can stay, because one of the issues that would make maybe make the mothers to abscond is the issues of where

they are staying if the wards are so congested. If they don't have proper beddings or proper beds, then that can lead to them thinking of absconding (self-discharging). But now with the proper wards and they are quite so roomy. . .we really do not experience many mothers trying to abscond." District health officer

## Reliance on families

Health workers recalled that though mothers and their families rarely refused KMC, they often wanted to go home after only a couple days, long before their baby had gained enough weight. Participants shared that the desire to go home stemmed from poor support at the hospital and a reliance on families for care and supplies, such as food, clothes, KMC wraps and hats to keep the baby warm. Some mentioned that the mother and her family were simply not expecting hospital admission after delivery and were unprepared for an extended stay, such as bringing only one cloth to wrap the newborn. Because the KMC wraps are not provided by the hospital, the family must arrange for more clothes to be brought alongside supplies for daily living needs such as food, clothes and bed linens, which was a burden for poor families served by these public hospitals. Faced with poor medical attention, including lack of review by clinicians and monitoring by nursing staff, as well as costs with maintaining her stay and concern about other children at home, health workers shared that women and their families frequently self-discharged early.

"Sometimes, it's because at this hospital we don't provide meals as such they complain that they don't have food so they run away. Others because they don't have a guardian they choose to run away. Others, it's due to family issues and you will just discover the other day they have absconded. But mostly it's due to having no guardians and no food. And also sometimes when they feel that the baby has gained a normal weight but they haven't been reviewed by health personnel sometimes they just decide to go." District nursing officer

"Actually those parents maybe they don't have enough things at the hospital let's say they don't have enough food, enough money to sustain them in the hospital, they will not concentrate on what we tell them and they will prefer taking the baby home, to take care for it at home. So that's the most important challenge that I have observed." District hospital clinical officer

Leaving early was also associated with misunderstandings and difficulties around KMC practice. Health workers shared that some mothers and caregivers agreed to initiate partly due to respect or fear of health care workers rather than their comfort or understanding of the practice. Health workers shared that sometimes staff did not have the time to fully explain. Due to low education and literacy rates among caregivers, health workers shared that it was difficult to explain KMC adequately. Misunderstandings and inadequate counselling contributed to difficulties in practices. Exacerbated by neglect in monitoring, these difficulties may not be adequately addressed by health workers.

On immediate acceptance of KMC: "I think it's out of fear. They ask themselves, "If we refuse, what next? I think it's because during that time mom is vulnerable and the baby is vulnerable and are always willing to accept." Tertiary hospital pediatrician

"A mother complained that she didn't get anything about putting the baby on body contact so she asked the nurse to explain everything again all together, but because of time the nurse didn't agree to repeat" District hospital nurse

Facilitators described by health workers included the availability of a caregiver to do KMC if the mother is unable to and support of husbands, mothers-in-law and grandparents, visitation rights for the mother's family and seeing peers practice KMC. However, because of a reliance of families to support care during KMC at the hospital, a lack of social support also discouraged mothers to continue. When the mother was alone with no one to help her, health workers shared that it magnified her feelings of depression and desire to go home.

"These guardians are supporting these mothers; there are always together. When the guardian wants to leave, leaving the mothers with the baby on kangaroo, the mother also wants to leave because they feel they have no support" District hospital nurse

## Discussion

Health workers from Malawi shared overall positive perspectives on KMC practice but highlighted a number of challenges to its implementation. While poor health condition of the mother and newborn delayed initiation of KMC, health workers reported that mothers and their families rarely refused KMC. However, a major challenge shared by health workers in our study was sustainability of practice and supporting mothers to continue facility-based KMC. Staff perceptions that KMC babies were stable may lead to neglect of the KMC ward. Neglect of the KMC ward by medical staff, inadequate staffing for adequate support and reliance on the mother and her social network for supplies were associated with women self-discharging before their baby had gained adequate weight.

Previously reported common implementation obstacles to routine uptake of KMC at health facilities include health worker perceptions that KMC is ineffective, that it is an inferior alternative to conventional incubators only suitable for low-resource health settings, that it increases workload and being hampered by restricted caregiver visitation times in neonatal intensive care units [9]. Beliefs in its efficacy and lack of clear guidelines was also found as a barrier to the adoption of facility-based KMC for nurses in LMICs [14]. In contrast, our research found that health workers supported KMC as effective, decreased workload, they were knowledgeable about KMC guidelines and there was a dedicated space for mothers to practice KMC without restriction. Health workers in our study focused on the challenge of inadequate staffing and overburdening of responsibilities, which has also been previously described as barriers to scaling up KMC in Africa, Asia and other LMICs settings [14, 15]. However, in expanding on human resource constraints reported broadly previously, health workers in our study further detailed its consequences in triaging priorities in care. Divisions of clinically stable and unstable infants prioritizing those in critical care was subsequently associated with neglect of the KMC ward and assumption of adequate monitoring by caregivers. This relates to what others have cautioned as the "empty scale-up" of KMC where conceptualization of KMC is limited to a unit or a place within the hospital, rather than part of the continuum of essential care for premature infants [16].

Our research with health workers in Malawian hospitals demonstrate the assertion that while KMC is a low-cost intervention, it is not a no-cost intervention [1]. Though KMC is described as a low-technology and low-cost solution suitable for resource-constrained health settings by both the literature on KMC and health workers themselves, it still needs investments in technology, staffing and hospital support in provisioning basic needs.

### Investment in technology

The promotion of KMC as a low-technology solution excludes newborns that are not clinically stable, a reported barrier to KMC initiation and associated with attitudes that KMC babies are

a lower priority for care. A number of studies about KMC implementation in sub-Saharan Africa, including from Ghana [17], Zambia [18], South Africa [19], and Uganda [20], have also noted the challenge of initiating KMC when newborns are receiving oxygen or phototherapy. There has been an effort to extend KMC practice to clinically unstable newborns, which was shown to be feasible and acceptable [21] and a number of trials are currently ongoing, including the WHO led i-KMC trials in Ghana, India, Malawi, Nigeria and Tanzania [22] and the OMWaNA trial in Uganda [23]. Investment in technologies, especially around monitoring equipment to support tracking vital signs, will be critical for clinically unstable newborns as well as for stable preterm and low birthweight infants between monitoring visits.

### Investment in staffing

Participants in this study highlighted the multiple roles that nurse midwives played from counselling caregivers on initiation to being the main medical provider monitoring infants during KMC as well as supporting labour, delivery and nursery wards during emergencies. The challenge of adequate staffing and workload especially during night, weekend and holiday shifts has been reported in studies in sub-Saharan Africa as a key barrier to KMC practice [17, 18, 21, 24–31]. Investments in staffing can help to move beyond basic practice of initiating and educating caregivers on danger signs to continued support. It strengthens capacity for and value of monitoring. Other studies on implementing KMC in sub-Saharan Africa identified maternal emotional and practical support from health workers as a key enabler for the practice [18, 19, 21, 30–34].

### Investment in basic needs

Study participants highlighted that having a dedicated KMC ward facilitated the practice, which is aligned with other studies in sub-Saharan Africa that described insufficient space as a critical challenge [19, 26, 28, 29, 35, 36]. However, beyond a sufficient space, there is a need for family-centered care to support mothers and their companions during their extended stay at the hospital. While KMC may be lower cost to the medical system than conventional care, there are considerable costs to the mother and her family. Investing in clothes, food, and entertainment and learning opportunities for mothers to help pass the time may help mothers and their families in their decision to stay and continue practicing KMC.

### Strengths and limitations

A strength of the study is that it elicited perspectives from a wide range of health workers in Malawian hospitals that were purposefully sampled. A limitation is that district and tertiary hospital interviews were not separated though they have different capacities. This is an area for future research. While the personal toll of facility-based KMC on families is described by health workers in our study, further research can explore caregiver perspectives and impacts of economic costs and isolation on maternal quality of life during extended hospitalizations for KMC.

## Conclusion

Health workers in our study viewed KMC positively and confirmed routine uptake of KMC as policy for stable low birthweight and preterm infants in the four hospitals. However, monitoring and maintenance of facility-based KMC practice were important challenges. In striving to end preventable neonatal deaths, strengthening hospital capacities to support KMC through

investments in technologies, staffing and provision of basic supplies is needed as part of a continuum of care for premature infants.

## Supporting information

**S1 File. COREQ checklist.**
(PDF)

**S2 File. Interview topic guide–English.**
(DOCX)

**S3 File. Interview topic guide–Chichewa.**
(DOCX)

## Acknowledgments

This manuscript is part of the "Integrating a neonatal healthcare package for Malawi" project within the Innovating for Maternal and Child Health in Africa (IMCHA) initiative. The authors are grateful to the IMCHA team for their support, the study participants for their voluntary participation, and the Directors of the various institutions included in the study for allowing us to conduct the study in their facilities.

## Author Contributions

**Conceptualization:** Laura Newberry, Jenala Njirammadzi, Tamanda Hiwa, Elizabeth M. Molyneux, Queen Dube, David M. Goldfarb, Kondwani Kawaza, Alinane Linda Nyondo-Mipando.

**Data curation:** Mai-Lei Woo Kinshella, Alinane Linda Nyondo-Mipando.

**Formal analysis:** Mai-Lei Woo Kinshella, Sangwani Salimu, Alinane Linda Nyondo-Mipando.

**Funding acquisition:** Queen Dube, David M. Goldfarb, Kondwani Kawaza.

**Investigation:** Mai-Lei Woo Kinshella, Sangwani Salimu, Brandina Chiwaya, Felix Chikoti, Lusungu Chirambo, Ephrida Mwaungulu, Tamanda Hiwa, Alinane Linda Nyondo-Mipando.

**Methodology:** Mai-Lei Woo Kinshella, Alinane Linda Nyondo-Mipando.

**Project administration:** Mwai Banda.

**Supervision:** Mai-Lei Woo Kinshella, Laura Newberry, Marianne Vidler, Elizabeth M. Molyneux, Queen Dube, Joseph Mfutso-Bengo, David M. Goldfarb, Kondwani Kawaza, Alinane Linda Nyondo-Mipando.

**Writing – original draft:** Mai-Lei Woo Kinshella.

**Writing – review & editing:** Mai-Lei Woo Kinshella, Sangwani Salimu, Brandina Chiwaya, Felix Chikoti, Lusungu Chirambo, Ephrida Mwaungulu, Mwai Banda, Laura Newberry, Jenala Njirammadzi, Tamanda Hiwa, Marianne Vidler, Elizabeth M. Molyneux, Queen Dube, Joseph Mfutso-Bengo, David M. Goldfarb, Kondwani Kawaza, Alinane Linda Nyondo-Mipando.

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
