## [Decision Letter · Decision Letter 0]

26 Oct 2020

PONE-D-20-31435

“So sometimes, it looks like it’s a neglected ward”: Health worker perspectives on implementing kangaroo mother care in southern Malawi

PLOS ONE

Dear Dr. Kinshella,

Thank you for submitting your manuscript to PLOS ONE. After careful consideration, we feel that it has merit but does not fully meet PLOS ONE’s publication criteria as it currently stands. Therefore, we invite you to submit a revised version of the manuscript that addresses the points raised during the review process.

There are only few comments and suggestions to improve the text. After this, the manuscript is ready to be accepted.

We look forward to receiving your revised manuscript.

Kind regards,

Ricardo Q. Gurgel, PhD

Academic Editor

PLOS ONE

Journal Requirements:

2. Please include a copy of the interview guide used in the study, in both the original language and English, as Supporting Information, or include a citation if it has been published previously.

4. Please amend the manuscript submission data (via Edit Submission) to include author Queen Dube.

Reviewers' comments:

Reviewer's Responses to Questions

**Comments to the Author**

1. Is the manuscript technically sound, and do the data support the conclusions?

Reviewer #1: Yes

Reviewer #2: Yes

Reviewer #3: Yes

2. Has the statistical analysis been performed appropriately and rigorously? 

Reviewer #1: N/A

Reviewer #2: I Don't Know

Reviewer #3: Yes

3. Have the authors made all data underlying the findings in their manuscript fully available?

Reviewer #1: Yes

Reviewer #2: Yes

Reviewer #3: Yes

4. Is the manuscript presented in an intelligible fashion and written in standard English?

Reviewer #1: Yes

Reviewer #2: Yes

Reviewer #3: Yes

5. Review Comments to the Author

Reviewer #1: NICELY written observations.

Please do refer to a recent publication in LANCET 2019 wherein community KMC was practised in rural settings.

Effect of community-initiated kangaroo mother care on survival of infants with low birthweight: a randomised controlled trial VOLUME 394, ISSUE 10210, P1724-1736, NOVEMBER 09, 2019

The observations made by nurses in KMC ward are strong and cannot be generalised on the basis of 27 people's views. Was 27 a convenient sample or calculated from some previous studies?

Observations like NEGLECTED WARD or DOCTOR to be dragged for visit once in 2-3days.. STRONG WORDS indeed which can push a hit button to opening of KMC wards in hospitals. ( KMC can prevent 50% neonatal death as mentioned in introduction)

In thematic analysis, most frequent coders are to be shown . What were those words of Family people?

Reviewer #2: This is a well-written manuscript that is reporting on important barriers to implementation and sustainability of KMC practice in LMICs. I have few suggestions to the authors that they need to address before acceptance:

1. Please submit the interview guide

2. More details are needed with respect to the way the interviews were analyzed. What theory was used (?grounded theory??), was it thematic analysis? Was the analysis entirely inductive or did you use some "a priori" themes from the pilot study? How did the pilot findings influence subsequent interviews/analysis? etc..

3. Minor edit, Abstract, L32: Data 'were' instead of 'was'. Data = pleural

4. Discussion: Can benefit from examples of success stories in countries with similar challenges, if available.

Best of luck

Reviewer #3: Thank you for the excellent manuscript. I learned a lot and the paper was well written. I would perhaps just make a few more notes about contrasts between low and high resource settings. Please see my comments on your paper.

6. PLOS authors have the option to publish the peer review history of their article (what does this mean?). If published, this will include your full peer review and any attached files.

Reviewer #1: **Yes: **Dr Surender Singh Bisht

Reviewer #2: **Yes: **Mona Nabulsi

Reviewer #3: No

---

## [Author Response · Author response to Decision Letter 0]

16 Nov 2020

Dear Dr. Gurgel and the editorial team,

Re: PONE-D-20-31435 “So sometimes, it looks like it’s a neglected ward”: Health worker perspectives on implementing kangaroo mother care in southern Malawi”

On behalf of the research team, we thank you for your thorough and constructive review of our manuscript. Please find below our responses to the queries raised with line numbers. 

Editorial comments: 

1. Please ensure that your manuscript meets PLOS ONE's style requirements, including those for file naming. The PLOS ONE style templates can be found at: 

Response: Thank you, we have cross-checked the PLOS ONE style templates and revised the manuscript to meet style requirements including headings formatting and supplementary file naming. 

2. Please include a copy of the interview guide used in the study, in both the original language and English, as Supporting Information, or include a citation if it has been published previously.

Response: Interview guides in both English and Chichewa have been added as supplementary information (S2 and S3).

3. We note that you have indicated that data from this study are available upon request. PLOS only allows data to be available upon request if there are legal or ethical restrictions on sharing data publicly. For information on unacceptable data access restrictions, please see http://journals.plos.org/plosone/s/data-availability#loc-unacceptable-data-access-restrictions. In your revised cover letter, please address the following prompts:

a. If there are ethical or legal restrictions on sharing a de-identified data set, please explain them in detail (e.g., data contain potentially identifying or sensitive patient information) and who has imposed them (e.g., an ethics committee). Please also provide contact information for a data access committee, ethics committee, or other institutional body to which data requests may be sent.

b. If there are no restrictions, please upload the minimal anonymized data set necessary to replicate your study findings as either Supporting Information files or to a stable, public repository and provide us with the relevant URLs, DOIs, or accession numbers. Please see http://www.bmj.com/content/340/bmj.c181.long for guidelines on how to de-identify and prepare clinical data for publication. For a list of acceptable repositories, please see http://journals.plos.org/plosone/s/data-availability#loc-recommended-repositories.

Response: Due to the nature of qualitative transcripts containing potentially directly and indirectly identifying data, the datasets analyzed during the current study are not publically available to protect participant confidentiality (ethic committees: University of Malawi College of Medicine (P.08/15/1783) and the University of British Columbia (H15-01463-A003)). For data inquiries, please contact COMREC Chairperson (comrec@medcol.mw).

4. Please amend the manuscript submission data (via Edit Submission) to include author Queen Dube.

Response: Thank you for your attention to detail. Queen Dube has been added to the manuscript data on the submission. 

Reviewer #1: 

1. NICELY written observations.

Response: Thank you, Dr Bisht, for your kind words. 

2. Please do refer to a recent publication in LANCET 2019 wherein community KMC was practised in rural settings.

Effect of community-initiated kangaroo mother care on survival of infants with low birthweight: a randomised controlled trial VOLUME 394, ISSUE 10210, P1724-1736, NOVEMBER 09, 2019

Response: Thank you for recommending the recent publication on community KMC. It is a very important area of research and fills an important gap in the current literature. However, Mazumder et al 2019 reports on an efficacy trial of community-based KMC in India while the objective of our study was to understand implementation factors related to facility-based KMC. Upon review of the manuscript, we found it difficult to find a meaningful line to add the reference due to different intervention contexts and study scopes. 

3. The observations made by nurses in KMC ward are strong and cannot be generalised on the basis of 27 people's views. Was 27 a convenient sample or calculated from some previous studies?

Response: Thank you for your comment. In line 101-105, we describe purposeful sample methodology and the cadres of health workers we sought in our study. In lines 106-108, we describe that a sample size of 5-10 per site was estimated to reach data saturation with a variety of perspectives due to the small size of the neonatal units. Each of the units in the district hospitals had only two to three nurses each. 

4. Observations like NEGLECTED WARD or DOCTOR to be dragged for visit once in 2-3days.. STRONG WORDS indeed which can push a hit button to opening of KMC wards in hospitals. ( KMC can prevent 50% neonatal death as mentioned in introduction)

Response: Thank you kindly for your observation. We also agree that it will push for opening KMC wards in hospitals but also for adequate skilled staffing, technology and supports for caregivers to make those KMC wards effective. 

5. In thematic analysis, most frequent coders are to be shown. What were those words of Family people?

Response: If we understand your question correctly, kindly see the “Reliance on families” sub-section in the results (starting on line 215). 

Reviewer #2: 

This is a well-written manuscript that is reporting on important barriers to implementation and sustainability of KMC practice in LMICs. I have few suggestions to the authors that they need to address before acceptance:

1. Please submit the interview guide

Response: Thank you, Prof Nabulsi, for your kind words and recommendations on strengthening the paper. We added the interview guide (S2 – in English, S3 – in Chichewa). 

2. More details are needed with respect to the way the interviews were analyzed. What theory was used (?grounded theory??), was it thematic analysis? Was the analysis entirely inductive or did you use some "a priori" themes from the pilot study? How did the pilot findings influence subsequent interviews/analysis? etc.

Response: More detail has been added, “We employed a thematic approach in analyzing the data. The interview guide was piloted with nurses at the central hospital to refine questions and preliminary analysis to develop the coding framework for analysis. The coding framework included training, initiation, monitoring, perceptions of KMC among health workers, health worker perceptions of caregiver attitudes and engagement, overall challenges and supports of KMC practice. Codes emerged inductively from the pilot data as well as deductively from our research questions” (lines 119-124).

3. Minor edit, Abstract, L32: Data 'were' instead of 'was'. Data = pleural

Response: Thank you for your attention to detail. This has now been corrected (see line 32).

4. Discussion: Can benefit from examples of success stories in countries with similar challenges, if available.

Response: Thank you for your comment, which has inspired a lot of thought within our team. In some ways, Malawi can be considered a success story for KMC because it was adopted as national policy over a decade ago and its practice was found in all four study hospitals. However, there continued to be challenges highlighting elements that worked and elements that required improvement. In line 317-319, we reference 7 other studies on implementing KMC in sub-Saharan Africa that identified maternal emotional and practical support from health workers as a successful element for KMC practice.

Reviewer #3:

Thank you for the excellent manuscript. I learned a lot and the paper was well written. I would perhaps just make a few more notes about contrasts between low and high resource settings. Please see my comments on your paper:

Abstract

1. What do you mean by this? (re: early recognition of a complication)

Response: Thank you for your kind words regarding the manuscript. Regarding early recognition of a complication, we have rephrased to “recognizing danger signs of illness” for clarity (see line 23). 

2. Can you clarify what you mean by basic supplies? 

Response: We added “such as food, bedding, and KMC wraps” to explain basic supplies (see line 45-46).

Introduction

3. Would add in low resource settings --in high resource settings KMC is not necessary for survival ,but important for development

Response: We thank reviewer 3 for their observation. Interestingly, one of the points discussed at the XIIth international conference on KMC recently published in 2020 (Charpak et al 2020) highlighted that KMC is no longer perceived as an alternative for the poor in low- and middle- income countries but a universal best-care practice for the survival and well-being of small and preterm infants in high-income countries as well. 

4. Clarify what you mean here (re: early recognition of any problems) 

Response: Like in the abstract, we rephrased to “recognizing danger signs of illness” for clarity (see line 60-61). 

Results

5. Can you explain what this word means? (re: absconding)

Response: Thank you for your comment. We added “self-discharging” in brackets to clarify (see line 212). 

6. This is so important ---and also is an issue in high resource settings, as family often isn't allowed in NICUs, and then the mother stays home and doesn't visit the baby. Can you discuss the parallels in the discussion? 

Response: Thank you very much for your comment. We added a line in the discussion to highlight this area as an important topic for future research: “While the personal toll of facility-based KMC on families is described by health workers in our study, further research can explore caregiver perspectives and impacts of economic costs and isolation on maternal quality of life during extended hospitalizations for KMC” (lines 333-336). 

Discussion

7. Do you happen to know what happens to these babies and mothers who self-discharge? Do they have a higher morbidity/mortality rate?

Response: Thank you for the great question. Unfortunately, we do not have this data. While we interviewed a broad spectrum of health workers on their experience implementing KMC in their facilities, we did not track individual outcomes of the KMC babies they recalled in their narratives. 

8. LOVE THIS INSIGHT!!! (re: In contrast, our research found that health workers supported KMC as effective, decreased workload, they were knowledgeable about KMC guidelines and there was dedicated space for mothers to practice KMC without restriction.)

Response: Thank you for your comment.

Please do not hesitate to contact us should you have any questions or areas that need clarification.

Best regards,

Mai-Lei Woo Kinshella, Sangwani Salimu, Brandina Chiwaya, Felix Chikoti, Lusungu Chirambo, Ephrida Mwaungulu, Mwai Banda, Laura Newberry, Jenala Njirammadzi, Tamanda Hiwa, Marianne Vidler, Elizabeth M Molyneux, Queen Dube, Joseph Mfutso-Bengo, David M. Goldfarb, Kondwani Kawaza

---

## [Editor Report · Decision Letter 1]

30 Nov 2020

“So sometimes, it looks like it’s a neglected ward”: Health worker perspectives on implementing kangaroo mother care in southern Malawi

PONE-D-20-31435R1

Dear Dr. Kinshella,

We’re pleased to inform you that your manuscript has been judged scientifically suitable for publication and will be formally accepted for publication once it meets all outstanding technical requirements.

Kind regards,

Ricardo Q. Gurgel, PhD

Academic Editor

PLOS ONE

---

## [Editor Report · Acceptance letter]

9 Dec 2020

PONE-D-20-31435R1 

“So sometimes, it looks like it’s a neglected ward”: Health worker perspectives on implementing kangaroo mother care in southern Malawi 

Dear Dr. Kinshella:

I'm pleased to inform you that your manuscript has been deemed suitable for publication in PLOS ONE. Congratulations! Your manuscript is now with our production department. 

Kind regards, 

on behalf of

Professor Ricardo Q. Gurgel 

Academic Editor

PLOS ONE